# Forest Area, CO₂ Emission, and COVID-19 Case-Fatality Rate: A Worldwide Ecological Study Using Spatial Regression Analysis

Hansen Li [1], Guodong Zhang [1,*] and Yang Cao [2,3,*]

[1] Institute of Sports Science, College of Physical Education, Southwest University, Chongqing 400715, China; hanson-swu@foxmail.com
[2] Clinical Epidemiology and Biostatistics, School of Medical Sciences, Örebro University, 70182 Örebro, Sweden
[3] Unit of Integrative Epidemiology, Institute of Environmental Medicine, Karolinska Institutet, 17177 Stockholm, Sweden
* Correspondence: yang.cao@oru.se (Y.C.); lygd777@swu.edu.cn (G.Z.); Tel.: +46-19-602-6236 (Y.C.); +86-157-3026-7257 (G.Z.)

**Abstract:** Spatial analysis is essential to understand the spreading of the COVID-19 pandemic. Due to numerous factors of multi-disciplines involved, the current pandemic is yet fully known. Hence, the current study aimed to expand the knowledge on the pandemic by exploring the roles of forests and $CO_2$ emission in the COVID-19 case-fatality rate (CFR) at the global level. Data were captured on the forest coverage rate and $CO_2$ emission per capita from 237 countries. Meanwhile, extra demographic and socioeconomic variables were also included to adjust for potential confounding. Associations between the forest coverage rate and $CO_2$ emission per capita and the COVID-19 CFR were assessed using spatial regression analysis, and the results were further stratified by country income levels. Although no distinct association between the COVID-19 CFR and forest coverage rate or $CO_2$ emission per capita was found worldwide, we found that a 10% increase in forest coverage rates was associated with a 2.37‰ (95%CI: 3.12, 1.62) decrease in COVID-19 CFRs in low-income countries; and a 10% increase in $CO_2$ emission per capita was associated with a 0.94‰ (95%CI: 1.46, 0.42) decrease in COVID-19 CFRs in low-middle-income countries. Since a strong correlation was observed between the $CO_2$ emission per capita and GDP per capita (r = 0.89), we replaced $CO_2$ emission with GDP and obtained similar results. Our findings suggest a higher forest coverage may be a protective factor in low-income countries, which may be related to their low urbanization levels and high forest accessibilities. On the other hand, $CO_2$ can be a surrogate of GDP, which may be a critical factor likely to decrease the COVID-19 CFR in lower-middle-income countries.

**Keywords:** COVID-19; forest; $CO_2$; fatality; death; mortality; health; nature

## 1. Introduction

At the end of 2019, a newly emerging virus, later known as SARS-CoV-2 or COVID-19, was spotted in China, which has unexpectedly started a global pandemic [1,2]. The novel virus is a sister clade to the previous SARS-CoV [3], less fatal [4] but more contagious [5]. As of February 2022, nearly 400 million confirmed cases and more than five million deaths were recorded globally [6], indicating the great impact of the pandemic on global public health.

In the past two years, numerous studies have been conducted on COVID-19, and those with spatial analysis may play an essential role in understanding the pandemic [7–9] because spatial spreading is a primary characteristic of all infectious diseases. Spatial methods have contributed to uncovering the relationships between the COVID-19 case-fatality rate (CFR) and sociodemographic or environmental factors, such as unemployment rate [10,11], commuting flows [12], and environmental pollution (e.g., $NO_2$ and particulate matter) [13,14]. Although many factors have been identified, the COVID-19 pandemic is still full of unknowns because various factors were involved, and their joint influence may

be complex [15]. Therefore, it is recommended to incorporate more variables of different kinds and disciplines to understand the COVID-19 phenomenon [16].

Forest is an ecological environment that provides mental and physical restoration resources during the COVID-19 pandemic [17], which may also help enhance the immune functions of visitors to fight against the disease [18]. Numerous studies have found negative associations between greenness and mortality rate [19–21]. Some recent studies have reported negative associations between green space coverage and COVID-19 CFRs, and air pollution reduction and immunoregulation are assumed to be potential reasons behind this [22–24]. However, most relevant studies did not focus on forests, and their data were only captured from one or several countries or regions. Therefore, the role of forests on the COVID-19 CFR requires further investigation.

$CO_2$ emission is a common ecological variable that is associated with climate change. During the COVID-19 pandemic, numerous studies were conducted concerning the role of harmful gas in the pandemic. For example, ozone, nitrogen dioxide, carbon monoxide, formaldehyde, and PM 2.5 have been reported to associate with higher COVID-19 infection or fatality rates [25,26]. In contrast, fewer studies have focused on the relationship between $CO_2$ emissions and COVID-19 CFRs, and the mechanisms are unclear [27,28]. $CO_2$ emission may affect health expenditure and is also a reflection of the social lockdown/blockade [29,30], therefore worth more attention.

Ecological studies collect and analyze data at the population level, which are convenient and inexpensive, and help identify ecological or environmental factors that are associated with infectious diseases. The aims of ecological studies include monitoring population health so that public health strategies may be developed, making large-scale comparisons between countries and regions, studying the relationship between population-level exposure to risk factors and health outcomes, or looking at the contextual effect of risk factors on the population [31]. Although other study designs are generally considered more reliable, particularly in the inference of causation, the population contexts have been shown to be stronger determinants of disease at the population level than individual-level risk factors [31]. Therefore, we aimed to examine the associations between COVID-CFRs and forests and $CO_2$ emissions worldwide while controlling socioeconomic factors' potential confounding using country-level data. Our study may explore ecological factors that may help monitor, predict, and mitigate the pandemic.

## 2. Materials and Methods

### 2.1. Data Sources

The data on the forest coverage rate by country were obtained from the World Bank [32]. The data providers include the Food Agriculture Organization, electronic files, and websites. Moreover, the data on the forest area per capita were also obtained from the Worldometer websites [33]. The data providers include the Food and Agriculture Organization of the United Nations, the United Nations Population Division, the United Nations Statistics Division, and the World Bank. The data on the $CO_2$ emission by country were obtained from the Emission Database for Global Atmospheric Research, European Commission [34]. Following the latest update released in October 2021, $CO_2$ emission data are now available for each country for the time period between 1970 and 2020.

The COVID-19 data included in this study were obtained from the Our World in Data website [35], a collection of the COVID-19 data maintained by the organization Our World in Data and updated daily. The dataset includes country-level daily data on confirmed cases, deaths, testing, and other variables of potential interest. There are 67 variables from 237 countries and territories in the dataset by 6 December 2021.

According to the ISO 3166-1 alpha-2 code for spatial modeling, the datasets were linked together using the global geospatial vector database. Variables with missing values in more than 70% of countries were excluded.

In total, 27 variables were included in the final analysis of the presented study. They are: country, country's geographic variables/information (shape, location, longitude, and

latitude of the centroid), forest coverage rate (in percentage of land area), forest area per capita, $CO_2$ emission per capita, total reported COVID-19 cases, total confirmed COVID-19 deaths, population, population density, gross domestic product (GDP) per capita, cardiovascular diseases (CVD) death rate, diabetes prevalence, stringency index of government's response to the COVID-19 pandemic (a composite measure based on 9 response indicators, including school closures, workplace closures, and travel bans, rescaled to a value from 0 to 100 with 100 = strictest response) [36], total tests for COVID-19, total vaccinations for COVID-19, life expectancy, median age of the population, percentage of population aged 65 years or older, percentage of population living in extreme poverty, percentage of female smokers, percentage of male smokers, hospital beds, percentage of population with basic handwashing facilities, and the human development index.

The study used data from publicly available worldwide databases, so it was not required and not possible to involve patients or the public in the design, conduct, reporting, or dissemination plans of our research. Hence, ethical approval is not applicable.

The study was conducted under the Guidelines for Accurate and Transparent Health Estimates Reporting (GATHER) statement [37].

### 2.2. Case-Fatality Rate (CFR)

The CFR of COVID-19 was defined as the number of total deaths due to COVID-19 divided by the number of confirmed COVID-19 cases by 6 December 2020 and multiplied by 1000. The CFR was selected for our study, as it may reflect disease severity and the efficiency of treatment and healthcare response, and strain [38]. Furthermore, the CFR is not constant, which can vary between populations and timeframes, and may depend on the interplay between the causative agent of the disease, the host, the environment, available treatments, and the quality of patient care. For instance, it can increase if the healthcare system collapses due to a sudden increase in cases [39]. The CFR in the current study is a crude rate without adjusting for age and sex due to no more detailed age/sex data available worldwide.

### 2.3. Statistical Analyses

Descriptive statistics (e.g., mean, standard deviation, median, minimum, maximum, and interquartile range (IQR)) were calculated for the variables.

Linear correlations between the variables were examined with Pearson's correlation coefficient [40]. Spatial autocorrelation (or dependence) is the relationship between spatial proximity in observational units and similarity among their values. Positive spatial autocorrelation refers to situations in which the nearer the observational units mean more similar values (and vice versa for its negative counterpart) [41]. This feature violates the assumption of independence among observations to be included in regression analyses. Spatial autocorrelation among the COVID-19 CFRs of the investigated countries was tested using a multivariate linear regression model and the Moran's I test [42]. The autocorrelation was visualized with the aid of the Matérn correlation coefficient [43].

Matérn correlation models were employed to check the relationship between the COVID-19 CFR and the ecological and socioeconomic variables. The latitude and longitude of the centroid of the countries were included as random effects in the Matérn correlation models [44].

Spearman's correlation was employed to probe for general patterns of associations within the core variables (forest, $CO_2$ emission, and COVID-19 CFR). Skewed variables were log-transformed before linear correlation or regression analysis. The multiple imputation method was employed to handle the missing values in the data (the missing values were assumed to be missing at random). A total of ten copies of the data were created, and each of the complete datasets was analyzed independently. Estimates of parameters of interest were then averaged across the ten copies to create a single estimate according to Rubin's rule [45].

Given the associations might differ by economic level, we also stratified the analyses by income levels of the countries defined by the World Bank [46].

The associations of the investigated ecological and socioeconomic variables with the COVID-19 CFR were presented as changes in the COVID-19 CFR per 10% increase in the ecological and socioeconomic variables. The corresponding 95% confidence intervals (CI) were also provided. An association with a *p*-value less than 0.05 was considered statistically significant.

All the analyses were performed in R V.4.1.2 (the R Foundation for Statistical Computing, Vienna, Austria) using the package spaMM [47], and in Python V.3.7 (Python Software Foundation) using the packages geopandas and geoplot [48].

## 3. Results

### 3.1. Descriptive Characteristics of the Variables

In total, 264,861,480 reported COVID-19 cases and 5,255,849 confirmed deaths related to COVID-19 between 1 January 2020 and 6 December 2021, from 194 countries were included in the study. The descriptive statistics of the variables are shown in Table 1.

**Table 1.** Descriptive statistics of the variables.

| Variable | N | Mean | SD | Median | Min | Max | IQR |
|---|---|---|---|---|---|---|---|
| Forest cover (%) | 207 | 32.41 | 23.98 | 31.23 | 0.01 | 97.41 | 38.74 |
| Forest area ($m^2$ per capita) | 202 | 11,055.09 | 28,442.80 | 2661.50 | 4.00 | 268,615.00 | 7561.25 |
| $CO_2$ emission (tons per capita) | 201 | 4.76 | 6.52 | 2.71 | 0.02 | 55.29 | 5.43 |
| Total COVID-19 cases | 194 | 1,370,420.43 | 4,789,589.59 | 194,817.50 | 1.00 | 49,085,361.00 | 713,475.25 |
| Total COVID-19 deaths | 186 | 28,257.25 | 89,303.97 | 2982.50 | 1.00 | 788,363.00 | 14,259.25 |
| COVID-19 CFR (‰) | 186 | 21.43 | 21.90 | 15.52 | 1.14 | 194.91 | 15.93 |
| Population (million) | 222 | 35.34 | 139.78 | 6.32 | 0.01 | 1444.22 | 22.64 |
| GDP per capita (USD) | 194 | 19,110.90 | 20,439.72 | 12,265.79 | 661.24 | 116,935.60 | 23,190.07 |
| Population density (per $km^2$) | 207 | 451.33 | 2089.29 | 87.18 | 0.14 | 20,546.77 | 176.53 |
| CVD death rate (per 100,000 people) | 189 | 264.92 | 122.70 | 244.66 | 79.37 | 724.42 | 164.06 |
| Diabetes prevalence (%) | 201 | 8.48 | 4.90 | 7.20 | 0.99 | 30.53 | 5.33 |
| Stringency index | 182 | 55.38 | 12.65 | 56.11 | 10.93 | 82.95 | 16.18 |
| Total tests for COVID-19 (per 1000 people) | 131 | 1340.29 | 2421.37 | 573.74 | 8.63 | 18,758.22 | 1229.37 |
| Total vaccinations for COVID-19 (per 100 people) | 217 | 96.76 | 60.73 | 101.75 | 0.01 | 297.99 | 98.62 |
| Life expectancy (year) | 218 | 73.38 | 7.49 | 74.71 | 53.28 | 86.75 | 10.64 |
| Median age of the population(year) | 190 | 30.30 | 9.12 | 29.50 | 15.10 | 48.20 | 16.65 |
| Percentage of population aged 65 years or older (%) | 188 | 8.61 | 6.12 | 6.22 | 1.14 | 27.05 | 10.42 |
| Percentage of population living in extreme poverty (%) | 125 | 13.88 | 20.25 | 2.50 | 0.10 | 77.60 | 20.80 |
| Percentage of female smokers (%) | 146 | 10.82 | 10.85 | 6.30 | 0.10 | 44.00 | 17.35 |
| Percentage of male smokers (%) | 144 | 32.90 | 13.67 | 32.25 | 7.70 | 78.10 | 19.00 |
| Hospital beds (per 1000 people) | 171 | 3.04 | 2.45 | 2.40 | 0.10 | 13.80 | 2.80 |
| Percentage of population with basic handwashing facilities (%) | 95 | 50.69 | 32.28 | 49.54 | 1.19 | 100.00 | 62.77 |
| Human development index | 189 | 0.72 | 0.15 | 0.74 | 0.39 | 0.96 | 0.23 |

Regarding the outcome and main factors of interests, we found a strong correlation between the COVID-19 CFR and $CO_2$ emission per capita (spearman $\rho = -0.37$, $p < 0.001$). However, we did not observe a distinct linear correlation between the COVID-19 CFR and forest coverage rate (spearman $\rho = 0.01$, $p = 0.859$) or forest area per capita (spearman $\rho = 0.07$, $p = 0.314$) (Figure 1).

In the linear correlation analysis for log-transformed variables (Supplemental Figure S1), high linear correlations were found between the forest coverage rate and forest area per capita ($r = 0.70$, $p < 0.001$); between $CO_2$ emission per capita and GDP per capita ($r = 0.89$, $p < 0.001$), the median age of the population ($r = 0.76$, $p < 0.001$), percentage of the population living in extreme poverty ($r = -0.83$, $p < 0.001$), percentage of the population with basic handwriting facilities ($r = 0.80$, $p > 0.001$), and the human development index ($r = 0.87$, $p < 0.001$); and between life expectancy and the percentage of the population aged 65 years or older ($r = 0.85$, $p < 0.001$), therefore, forest area per capita, GDP per capita, median age of the population, percentage of population aged 65 years or older, percentage of the population living in extreme poverty, percentage of the population with basic handwriting facilities, and the human development index were excluded in later regression analysis.

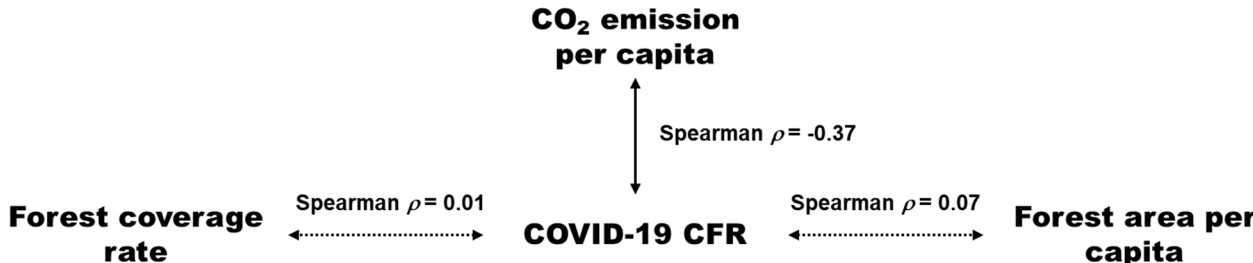

**Figure 1.** The relationships between forest coverage rage, $CO_2$ emission per capita, and COVID-19 CFR. Note: the solid line indicates significant linear correlation ($p < 0.05$); the dotted lines indicate non-significant linear correlation ($p > 0.05$).

### 3.2. Worldwide Distributions of Forest, $CO_2$ Emission, and COVID-19 CFR

Distributions of forest coverage rate and area per capita, $CO_2$ emission per capita, and COVID-19 CFRs of the countries with data available are shown in Figure 2. The highest values for the forest coverage rate and area per capita were found in North America, northern Europe, northern South America, and southern Africa; for $CO_2$ emissions per capita, in North America, Western Europe, the Middle East, and Australia; for the COVID-19 CFR, in southern North America, western South America, northern Africa, and eastern Asia.

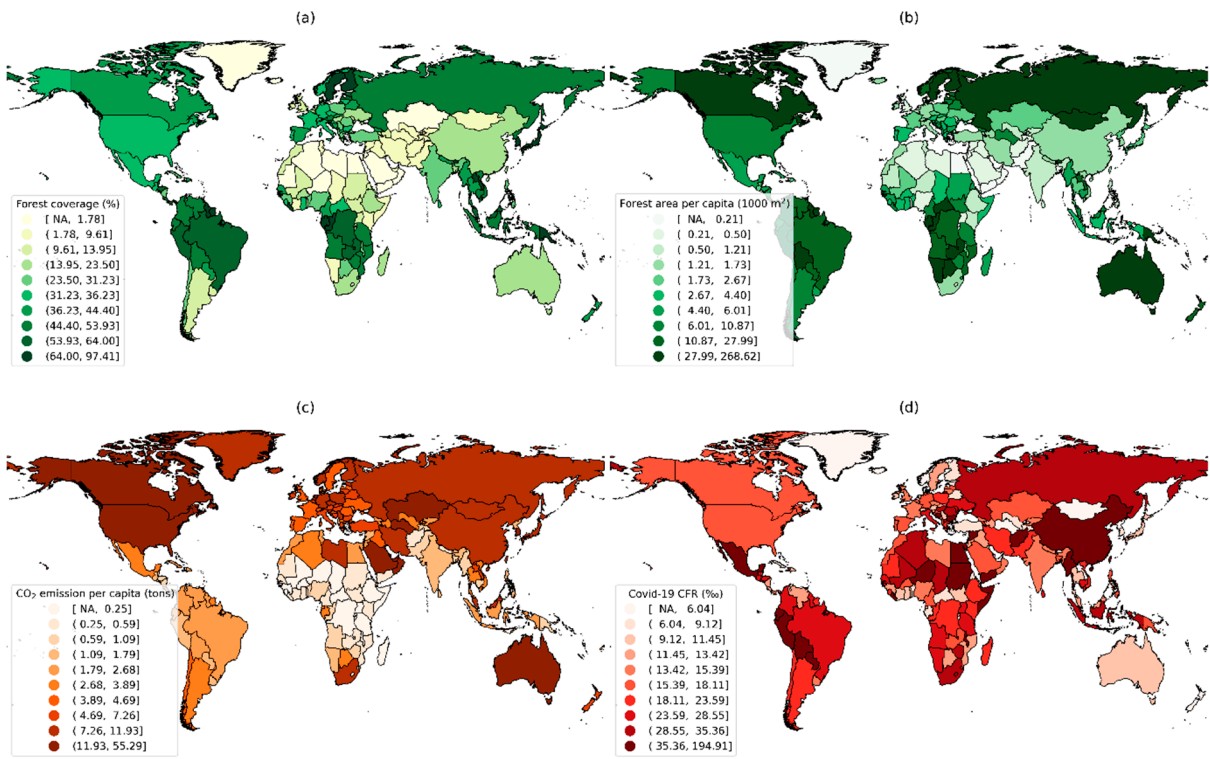

**Figure 2.** Global distribution of (**a**) forest coverage rate, (**b**) forest area per capita, (**c**) $CO_2$ emission per capita, and (**d**) COVID-19 CFR. CFR: case-fatality rate. NA: not available.

### 3.3. Spatial Autocorrelation of the COVID-19 CFR

The residuals of the COVID-19 CFR from the common (non-spatial) multivariate linear regression model show apparent spatial dependence among the countries/territories (Figure 3). The $p$-value from Moran's I test for the spatial autocorrelation of the residuals is 0.021.

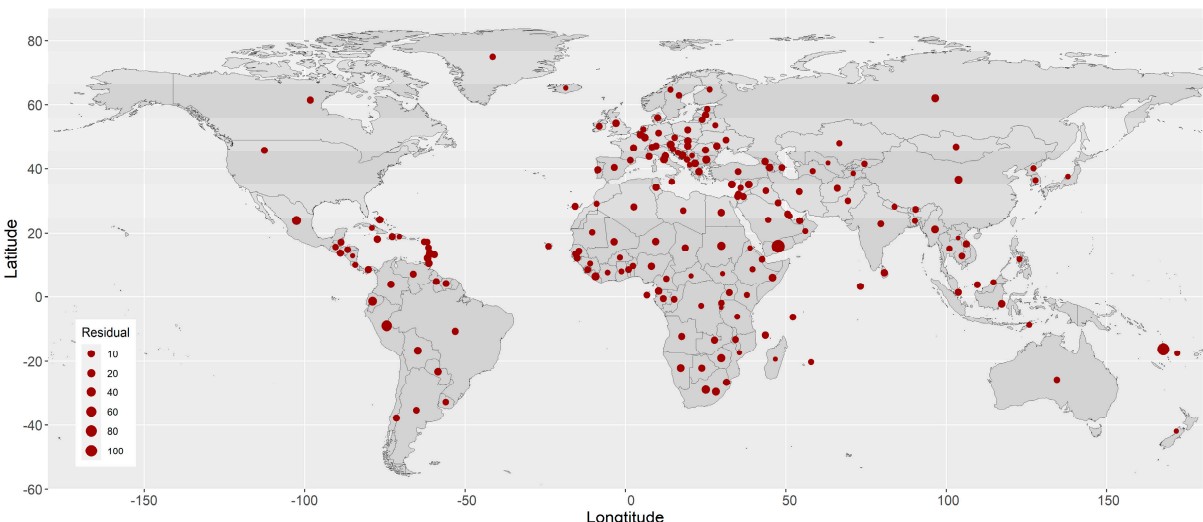

**Figure 3.** Residuals from the common (non-spatial) multivariate linear regression model.

The estimated spatial autocorrelation coefficient of the COVID-19 CFR between two locations against their distance is shown in Supplemental Figure S2, with a strength parameter $\nu = 10.33$ and a decay parameter $\rho = 0.89$. Basically, the autocorrelation coefficient is smaller than 0.5 if the distance between a pair of locations is larger than 6.2° in terms of the difference between their longitudes or latitudes. The ranges of longitude and latitude are $(-177.1508°, 178.5574°)$ and $(-54.4881°, 78.8286°)$, respectively. (Supplemental Figure S2).

### 3.4. Association of Ecological and Socioeconomic Variables with COVID-19 CFR

After controlling for the spatial dependence, the multivariable-adjusted associations of the investigated ecological variables (i.e., forest coverage rate and $CO_2$ emission per capita) and socioeconomic variables are presented as changes in the COVID-19 CFR per 10% increase in the variables in Table 2.

Worldwide, no statistically significant associations of forest coverage rate or $CO_2$ emission per capita with COVID-19 CFRs were found. However, the stratified results by economic level of the countries indicate that per 10% increase in forest coverage rate was associated with a 2.37‰ (95%CI: 1.62–3.12‰) decrease in the COVID-19 CFR in low-income countries, and per 10% increase in $CO_2$ emission per capita was associated with a 0.94‰ (95%CI: 0.42–1.46‰) decrease in the COVID-19 CFR in lower-middle-income countries (Figure 4).

Regarding the socioeconomic variables, life expectancy and the CVD death rate were associated with 0.12‰ (95%CI: 0.04–0.20‰) and 1.05‰ (95%CI: 0.12–1.98‰) increases in the COVID-19 CFR, respectively, and the stringency index was associated with a 0.80‰ (95%CI: 0.51–1.09‰) decrease in the COVID-19 CFR worldwide (Table 2).

In low-income countries, besides the forest coverage rate, female smokers and COVID-19 vaccinations were associated with increased (2.14‰: 95%CI: 1.64–2.65‰) and decreased (−2.37‰; −3.12−−1.62‰) COVID-19 CFRs, respectively.

In lower-middle-income countries, total population and population density were associated with decreased (−0.94‰: 95%CI: −1.46−−0.42‰) and increased (0.33‰; 0.08–0.57‰) COVID-19 CFRs, respectively.

In upper-middle-income countries, only the stringency index was associated with a decreased (−0.95‰: 95%CI: −1.50−−0.41‰) COVID-19 CFR.

In high-income countries, female smokers and CVD death rates were associated with increased COVID-19 CFRs, with increments of 0.32‰ (95%CI: 0.12–0.51‰) and 1.01‰ (95%CI: 0.56–1.49‰), respectively; and the stringency index was associated with a decreased (−0.36‰ (95%CI: −0.49–0.23‰) COVID-19 CFR.

**Table 2.** Changes in COVID-19 CFR (‰) per 10% increase in the ecological and socioeconomic variables.

| Variables | All Countries (N = 186) | | Low-Income Countries (N = 26) | | Lower-Middle-Income Countries (N = 50) | | Upper-Middle-Income Countries (N = 51) | | High-Income-Countries (N = 59) | |
|---|---|---|---|---|---|---|---|---|---|---|
| | CFR Change (95% CI) | *p* | CFR Change (95% CI) | *p* | CFR Change (95% CI) | *p* | CFR Change (95% CI) | *p* | CFR Change (95% CI) | *p* |
| Forest cover | −0.16 (−0.37, 0.06) | 0.198 | **−2.37 (−3.12, −1.62)** | **0.003** | 0.09 (−0.18, 0.36) | 0.557 | 0.17 (−0.20, 0.53) | 0.426 | 0.05 (−0.02, 0.13) | 0.215 |
| $CO_2$ emission per capita | −0.24 (−0.61, 0.13) | 0.215 | 0.93 (−0.02, 1.87) | 0.120 | **−0.94 (−1.46, −0.42)** | **0.004** | 0.13 (−0.77, 1.04) | 0.646 | −0.13 (−0.34, 0.08) | 0.274 |
| Population | 0.04 (−0.11, 0.18) | 0.635 | 0.46 (−0.32, 1.23) | 0.173 | **0.33 (0.08, 0.57)** | **0.030** | 0.17 (−0.13, 0.46) | 0.308 | 0.02 (−0.04, 0.08) | 0.397 |
| Population density | −0.10 (−0.31, 0.11) | 0.359 | −1.89 (−2.82, −0.95) | 0.085 | −0.31 (−0.65, 0.02) | 0.135 | −0.21 (−0.56, 0.14) | 0.306 | 0.03 (−0.05, 0.11) | 0.553 |
| Life expectancy | **0.12 (0.04, 0.20)** | **0.009** | 0.36 (0.18, 0.53) | 0.121 | 0.09 (0, 0.19) | 0.068 | 0.11 (−0.03, 0.24) | 0.155 | 0.02 (−0.03, 0.08) | 0.437 |
| Male smoker | −0.01 (−0.03, 0.02) | 0.522 | −0.05 (−0.15, 0.04) | 0.189 | 0 (−0.02, 0.02) | 0.485 | 0.00 (−0.04, 0.04) | 0.650 | −0.01 (−0.03, 0.00) | 0.118 |
| Female smoker | 0.25 (−0.02, 0.52) | 0.112 | **2.14 (1.64, 2.65)** | **0.002** | 0.11 (−0.27, 0.5) | 0.312 | 0.07 (−0.32, 0.45) | 0.578 | **0.32 (0.12, 0.51)** | **0.018** |
| CVD death rate | **1.05 (0.12, 1.98)** | **0.034** | −0.30 (−3.86, 3.25) | 0.228 | −0.64 (−1.77, 0.49) | 0.364 | −0.65 (−2.01, 0.71) | 0.390 | **1.03 (0.56, 1.49)** | **<0.001** |
| Diabetes prevalence | 0.31 (−0.32, 0.93) | 0.354 | 0.33 (−0.89, 1.56) | 0.150 | −0.12 (−0.63, 0.4) | 0.571 | −0.40 (−1.9, 1.09) | 0.605 | 0.19 (−0.17, 0.55) | 0.355 |
| Hospital beds | −0.17 (−0.64, 0.30) | 0.458 | 2.07 (0.61, 3.54) | 0.107 | 0.1 (−0.5, 0.7) | 0.548 | −0.09 (−1.06, 0.87) | 0.670 | 0.17 (−0.11, 0.45) | 0.343 |
| Stringency index | **−0.80 (−1.09, −0.51)** | **<0.001** | −0.60 (−1.54, 0.33) | 0.231 | −0.47 (−0.84, −0.1) | 0.087 | **−0.95 (−1.50, −0.41)** | **0.003** | **−0.36 (−0.49, −0.23)** | **<0.001** |
| Tests | 0.24 (−0.06, 0.54) | 0.151 | −0.23 (−0.83, 0.36) | 0.101 | 0.2 (−0.2, 0.6) | 0.357 | −0.58 (−1.40, 0.24) | 0.177 | 0.07 (−0.54, 0.68) | 0.484 |
| Vaccinations | −0.16 (−0.37, 0.06) | 0.198 | **−2.37 (−3.12, −1.62)** | **0.003** | 0.09 (−0.18, 0.36) | 0.557 | 0.17 (−0.20, 0.53) | 0.426 | 0.05 (−0.02, 0.13) | 0.215 |

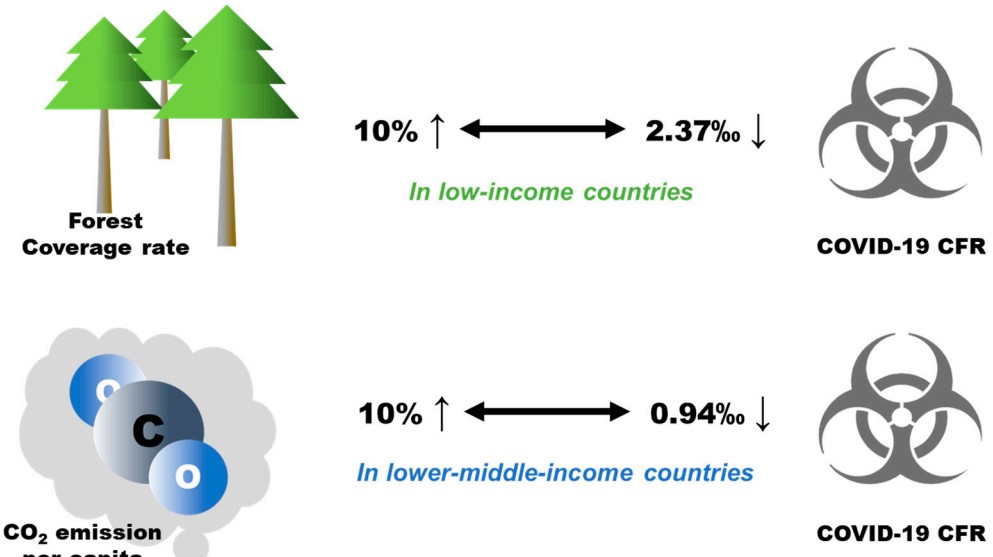

**Figure 4.** The stratified analyses on associations between forest coverage rate, $CO_2$ emission per capita, and COVID-19 CFR.

## 4. Discussion

This study examined the association between COVID-19 CFRs and forest or $CO_2$. Although no general patterns were observed at the global level, some notable results were observed in specific countries. Our findings imply the indicative functions of forests and $CO_2$ emission in the global pandemic. We could not provide solid mechanisms for explaining the observed phenomenon due to our study design. Nevertheless, we have tried to propose some potential clues or reasons to help understand the findings, which are presented in the following sections.

### 4.1. Forest Coverage and COVID-19 CFR

The current study included forest coverage rate and forest area per capita to indicate forest resources. The global analysis did not reveal a statistically significant association between the COVID-19 CFR and any of the forest variables. This result is inconsistent with a previous study where a dose-response association was spotted between green space and a reduced risk of COVID-19 mortality among counties in the USA from January to July 2020 [23]. However, another study reported that greenness was not associated with COVID-19 mortality in all counties in the USA [49]. These differences may come from different measures for the "natural environment" (leaf area index for the previous study), statistical methods, and spatiotemporal frameworks employed for analysis. The null effect we observed indicates that the role of and measures for "exposure to nature" should be considered cautiously. Currently, global urbanization has gathered most of the population into cities [50]. However, most forest areas are outside cities, especially in highly urbanized countries. Since distance strongly affects green space utilization [51,52], urban green spaces within or around cities may be the most popular and accessible type of natural environment for urban residents during the pandemic [53]. Thus, the forest variables calculated based on national forest area may not necessarily indicate the actual utilization or exposure to natural environments among residents in urbanized countries.

In the analysis stratified by income, we found that both the forest coverage rate and forest area per capita (Supplemental Table S1) were associated with decreased COVID-19 CFRs in low-income countries. Previous studies have speculated on some mechanisms to understand the decreased COVID-19 CFR, such as pollution reduction, physical activity promotion, and immunoregulation induced by microorganisms in natural environments [22,23,54]. However, these mechanisms did not consider low-income coun-

tries, and they cannot explain the discrepancies among countries with different income levels. As discussed above, the accessibility of forests may be a problem in urbanized countries. In contrast, those low-income countries are less urbanized due to being highly reliant on agriculture [55], and more residents are distributed in rural areas, such as living in small villages and small towns. Therefore, the residents in low-income countries may live closer to natural environments than residents in urbanized countries. According to Pichlerová, M., D. Önkal, A. Bartlett, J. Výbošťok, and V. Pichler [17], forest coverage rate and settlement size are both factors of forest accessibility. A higher forest coverage and a smaller settlement size both indicate more forest visits. Therefore, for residents scattered in rural areas, a higher forest coverage rate may imply a greater chance of forest visits. In addition to physically entering forests, forests around neighborhoods may also benefit them remotely. For example, a recent study in Italy found that the percentage of COVID-19 deaths was lower in greener areas when compared to the regions with low forest coverage [56]. The phenomenon, explained by the author, might partially result from non-deciduous Mediterranean plants, which emit volatile immunomodulatory and antiviral compounds. These volatile biogenic compounds are likely to benefit local residents and reduce COVID-19 CFRs [18]. According to the classification of the World Bank, most of the low-income countries are in tropical or subtropical Africa, where the climate is mild, and evergreen forests play a dominant role. Therefore, residents in low-income countries may have more chances to be affected by forests in their residential neighborhoods.

*4.2. $CO_2$ Emission and COVID-19 CFR*

We did not observe a distinct association between $CO_2$ emission per capita and COVID-19 CFRs worldwide, which is inconsistent with a recent study, where data from 119 countries or regions indicated a nonlinear relationship between the CFR and $CO_2$ emission [27]. Some points may explain the inconsistency, such as the differences in countries or regions included, dates of data access, and the measures for $CO_2$ emission (per capita vs. per country). Nevertheless, our results may, to a certain extent, challenge the view that $CO_2$ emission is a critical factor likely to increase total coronavirus cases and death rates [27].

Aside from being a common environmental indicator [57], $CO_2$ emission may somewhat reflect economic conditions, as economic growth mostly depends on the consumption of energy resources [58]. For example, many studies have suggested the potential relationship between $CO_2$ emissions and GDP [59]. A recent study on COVID-19 has underlined the method of estimating the changes in $CO_2$ emission with the aid of GDP when detailed energy activity data are unavailable [60]. In the current study, we also observed a strong positive correlation between the GDP per capita and $CO_2$ emission (r = 0.89), implying that $CO_2$ emission may be a surrogate of GDP. For this reason, we conducted a sensitivity analysis, which includes GDP (per capita) instead of $CO_2$ emission in the model. We did not observe a distinct association between the GDP per capita and COVID-19 CFR in the global analysis either (Supplemental Table S2), suggesting that GDP might not be a critical factor associated with the COVID-19 CFR worldwide. This finding appears counter-intuitive, as GDP is associated with financial resources for health protection [29]. However, we should realize that pressures on medical and public health services due to COVID-19 varied among countries. For well-developed countries, such as Japan, healthcare services were affordable, and the medical system has remained functional throughout [61]. Therefore, other factors may play dominant roles in well-developed countries, such as age structure [62,63], which may mask the role of GDP in fighting the COVID-19 pandemic globally.

In the analysis stratified by income level, we found that $CO_2$ emission was associated with a decreased COVID-19 CFR in lower-middle-income countries, and the association remained statistically significant in the sensitivity analysis when $CO_2$ emission was replaced with GDP per capita (Supplemental Tables S2 and S3). This result is not surprising, given the reasons mentioned above. Compared to upper-middle and high-income countries, where medical and infrastructure resources are usually sufficient [64], lower-middle-income countries are more vulnerable to the pandemic [65], especially developing countries with

poor economic conditions [66]. Therefore, a better economic condition may be more critical for lower-middle-income countries to fight against the pandemic.

In addition, the association between GDP and the COVID-19 CFR in low-income countries was ambiguous when the model was loaded with different forest variables (Supplemental Tables S2 and S3). We cannot conclude if GDP is an essential factor for low-income countries. Theoretically, they may face a similar crisis to lower-middle-income countries because resource shortage has also been reported as a common issue in low-income countries [64]. Further studies are warranted to check their differences.

### 4.3. Limitations

There are several notable limitations to the current study. First, to indicate the typical natural environment worldwide, we used forest coverage rate and forest area per capita in the analysis. As discussed above, these measures may not necessarily indicate a potential exposure to nature. There is not a satisfactory measure to indicate the actual exposure to natural environments at a worldwide level. Even if the results were limited to the residential areas, common measures such as the normalized difference vegetation index or enhanced vegetation index may not explore the properties of natural environments, as well as their qualities, accessibility, and availability [67]. Future studies may refer to the data of population mobility, such as location reports of smartphone applications, and incorporate it with green space indicators to quantify actual green space usage.

Second, there is no globally accepted definition of a COVID-19-related death. Thus, reporting bias cannot be ruled out. Moreover, given the fact that medical standards can vary with regions, it is not clear if some low-income countries can effectively report COVID-19-related deaths. Thus, their CFR values may be underestimated.

Third, though we tried to access the latest data worldwide, there are some discrepancies in their timeframes. Due to the impact of COVID-19, the patterns of $CO_2$ emissions may be changed with time and region. Therefore, the identified relationships may need re-examination in further work.

### 5. Conclusions

This study aimed to examine the roles of forest and $CO_2$ emission on COVID-19 CFRs while controlling known confounders. We conducted a spatial analysis that included worldwide-level data obtained from public access databases. Although we did not observe a distinct association between the COVID-19 CFR and forest coverage rate or forest area per capita worldwide, we found that the forest coverage rate and forest area per capita were associated with lower COVID-19 CFRs in low-income countries, which is preliminarily speculated to be related to their lower urbanized levels.

On the other hand, we did not find a distinct association between $CO_2$ emissions per capita and COVID-19 CFRs at the worldwide level either. Nevertheless, we found that $CO_2$ emissions per capita were associated with lower CFRs in lower-middle-income countries. Notably, we assumed that $CO_2$ emission might be a surrogate of GDP as they showed a strong correlation. Similar results were obtained when we replaced the $CO_2$ emission per capita with GDP per capita in the regression. These findings suggest that economic conditions may be a critical factor for lower-middle-income countries to fight against the current pandemic.

Generally, our findings suggest that forest and $CO_2$ emissions are two potential ecological factors that may help to monitor and predict the global pandemic. Given the limitation of the public access data, further research is warranted to uncover the mechanisms behind the associations.

**Supplementary Materials:** The following supporting information can be downloaded at: https://www.mdpi.com/article/10.3390/f13050736/s1, Figure S1: Pairwise scatter plots and Pearson's correlation coefficients of the independent variables; Figure S2: Strength and decay of the spatial autocorrelation between pair of locations: Table S1: Changes in COVID-19 CFR (‰) per 10% increase in the ecological and socioeconomic variables (including forest area per capita instead of forest cover);

Table S2: Changes in COVID-19 CFR (‰) per 10% increase in the ecological and socioeconomic variables (including GDP per capita instead of $CO_2$ emission per capita); Table S3: Changes in COVID-19 CFR (‰) per 10% increase in the ecological and socioeconomic variables (including GDP per capita instead of $CO_2$ emission per capita; and forest area per capita instead of forest cover).

**Author Contributions:** Conceptualization, Y.C.; methodology, Y.C.; data curation, Y.C.; writing—original draft preparation, Y.C. and H.L.; writing—review and editing, Y.C. and H.L.; supervision, Y.C. and G.Z.; project administration, Y.C. All authors have read and agreed to the published version of the manuscript.

**Funding:** This research received no external funding.

**Conflicts of Interest:** The authors declare no conflict of interest.

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
