# Peer review of "Forest Area, CO2 Emission, and COVID-19 Case-Fatality Rate: A Worldwide Ecological Study Using Spatial Regression Analysis"

_forests, doi:10.3390/f13050736_

Round 1
Reviewer 1 Report
In this paper, authors assessed the relationships between the role of forests, CO2 emission, and COVID-19 case-fatality rate.
The researchers accomplished this objective through evaluating data from 237 countries via spatial regression analysis. Their findings suggest that an increase in forest coverage rate was associated with a decrease in COVID case-fatality rate in low- and low-middle-income countries.
This manuscript is timely and would provide a valuable discussion on the relationships between the presence of forest systems and level of development with important health criteria.
The manuscript does need to be read carefully and reviewed for grammar.
Nevertheless, I found this manuscript interesting and well-organized. I specifically appreciate the global scope of the study. I would like to see this manuscript published in Forests once the authors address the following general comments. I have included more comments throughout the manuscript itself.
- I recommend being consistent with how you write “CO2.” Instances before line 76 included the “2” as a sub-script.
- I suggest re-formatting Table 1 to include a label for the variable column and left-justify the variable column (Column 1).
- I recommend you edit the sentences in lines 244-246 and 354-356 for clarity.

Reviewer 2 Report
In the paper, the authors explored different statistical approaches to evaluate the role of forests and carbon dioxide emissions in impacting the COVID pandemic. The study is very relevant and interesting which utilizes the applications of diverse data portals like organizations, websites, surveys, socio-economic reports, and many more for a spatially distributed ecological analysis connecting COVID rates and environmental factors. The study exhibits a good quality of scientific writing and presentation. Also, the intent and methodology of the study look interesting. However, the sections such as ‘3. Results’, should be improved as per the below mentioned comments. According to me, the manuscript can be accepted for publication after minor revisions.
As a reviewer, I pointed out some of the areas of the work where some revision needs to be introduced for improving the manuscript.
- Section ‘1. Introduction’ may be enriched with more References between lines 39 and 48.
- The major outcome or emphasis of Section ‘3.3. Spatial autocorrelation of the COVID-19 CFR’ is unclear. Please include what the results of this section infer.
- What do you mean by “strength parameter ν = 10.33 and a decay parameter ρ = 0.89. Basically, locations more than 6.2° (in longitude or latitude) away have an autocorrelation coefficient below 0.5”. What are the ranges of these variables? Please clarify.
- Figure 3 can be moved to Supplementary Material.
- It is recommended to have figures that show the interconnections between COVID rates and the environmental factors (CFR vs. forest cover, CFR vs. CO2 emissions).
- It is recommended to have a figure that shows the positive impact of forest cover on CFR and the negative impact of CO2 emissions on CFR such that the main intent of the study is well demonstrated.
- Section ‘5. Discussion’ is well written and can potentially contribute to vital future research.
- Section ‘5. Conclusions’ is too brief. Please expand this section by incorporating the main goal of the study, main methods used, main findings from the study with shreds of evidence, and the major contribution of the study.
As a reviewer, I recommend accepting the manuscript in the journal, Forests after the revisions are incorporated.
